# Chemistry and Analysis of Organic Compounds in Dinosaurs

**DOI:** 10.3390/biology11050670

**Published:** 2022-04-27

**Authors:** Mariam Tahoun, Marianne Engeser, Vigneshwaran Namasivayam, Paul Martin Sander, Christa E. Müller

**Affiliations:** 1PharmaCenter Bonn, Pharmaceutical Institute, Pharmaceutical & Medicinal Chemistry, University of Bonn, D-53121 Bonn, Germany; mtahoun@uni-bonn.de (M.T.); vnamasiv@uni-bonn.de (V.N.); 2Kekulé Institute for Organic Chemistry and Biochemistry, University of Bonn, D-53121 Bonn, Germany; marianne.engeser@uni-bonn.de; 3Institute of Geosciences, Section Paleontology, University of Bonn, D-53113 Bonn, Germany; martin.sander@uni-bonn.de

**Keywords:** fossil, dinosaur, molecular paleontology, paleoproteomics, porphyrin, collagen, melanin, keratin

## Abstract

**Simple Summary:**

Fossils of dinosaurs other than birds are at least 66 million years old. Nevertheless, many organic compounds have survived fossilization and can still be found in the fossils. This article describes the discovery of organic molecules in dinosaur fossils. It provides a review of the analytical methods used for their detection and characterization, and presents the wide range of chemical organic compounds, including small molecules and polymers, that have been found in dinosaurs to date. The difficulties in unambiguously confirming the presence of some of the organic molecules in these fossils are also discussed.

**Abstract:**

This review provides an overview of organic compounds detected in non-avian dinosaur fossils to date. This was enabled by the development of sensitive analytical techniques. Non-destructive methods and procedures restricted to the sample surface, e.g., light and electron microscopy, infrared (IR) and Raman spectroscopy, as well as more invasive approaches including liquid chromatography coupled to tandem mass spectrometry (LC-MS/MS), time-of-flight secondary ion mass spectrometry, and immunological methods were employed. Organic compounds detected in samples of dinosaur fossils include pigments (heme, biliverdin, protoporphyrin IX, melanin), and proteins, such as collagens and keratins. The origin and nature of the observed protein signals is, however, in some cases, controversially discussed. Molecular taphonomy approaches can support the development of suitable analytical methods to confirm reported findings and to identify further organic compounds in dinosaur and other fossils in the future. The chemical properties of the various organic compounds detected in dinosaurs, and the techniques utilized for the identification and analysis of each of the compounds will be discussed.

## 1. Introduction

After an organism’s death, microbial decomposition of organic constituents occurs very fast, mostly leaving behind mineralized skeletal remains. If this degradation process is arrested early enough, due to factors related to the burial environment and dependent on the characteristics of the molecular or tissue components [1,2,3], preservation of “soft tissue” can occur. Such fossils are exceptional and very valuable because they may contain information related to evolution, biology, or the environment that can be revealed by analyzing their composition [2,3]. Preserved soft tissue has been reported from a variety of fossil fish, amphibians, reptiles, dinosaurs, and mammals. This includes cells, organelles, skin, scales, feathers, hair, colored structures, digestive organs, eggshells, and muscles [2]. This mode of preservation is unique because the original organic material is minimally altered. Furthermore, this should be differentiated from fossilization involving alteration of original material, e.g., replacement of organic matter by minerals such as phosphates (phosphatization) or conversion to thin films of carbon (carbonization) [4,5].

Researchers try to understand the factors that hinder decay processes and contribute to the preservation of organic compounds present in soft tissues. These include (but are not limited to) intrinsic properties of the organic molecules, their environment (including metals such as Fe and Mn present) [6,7,8], and the type of preserved soft tissue. The presence of moisture, microorganisms or enzymes speeds up the decay process [1]. The most labile molecular bonds are first targeted during decomposition. For example, proteins and DNA are susceptible to degradation by hydrolysis of their peptide and phosphoric acid ester bonds, respectively. However, association of organic compounds with minerals (e.g., in bone or teeth) or with macromolecules may isolate and protect them from the external environment. 

Oxidative conditions usually lead to faster decay than reductive conditions. Hydrophobic organic compounds are more likely to be preserved than hydrophilic compounds because of their limited water-solubility, which protects them from hydrolysis and other reactions. Polymeric structures may be preserved due to crosslinking and intramolecular interactions. Environmental factors greatly affect fossilization, e.g., by applying pressure on tissue, limiting the mobility of molecules and exposure to water, microbes, and enzymes. Moreover, extremes of temperature, pH, and salinity play a role in molecular preservation by inhibiting microbial activity and affecting the rate of the chemical decomposition process. Taphonomic studies at a molecular scale (“molecular taphonomy”) can be used to establish analytical methods for understanding chemical processes that lead to the degradation of organic compounds upon fossilization ([1,2,9,10,11,12,13,14,15] and references therein).

Since the first discoveries of microstructures (collagen-like fibrils, vessels, and cells) in a 200-million-year-old dinosaur bone in 1966 [16] there has been an increased interest in studying the large number of available dinosaur fossils for signs of molecular preservation of organic compounds. Such finds provide information about the dinosaurs’ biology, including their evolution, eating habits, and environment. Most reports on organic matter in fossilized dinosaurs have been focused on their bones. In recent years, studies on eggshells, cartilage, feathers, and integumentary structures have emerged, albeit mostly discussing in situ analyses, and relying on morphological and microscopic observations due the uniqueness of the studied fossils (reviewed in [2,10] and references therein). 

To date, organic compounds have been recovered from a wide array of dinosaur taxa, including the early-branching coelurosaur *Sinosauropteryx* [17], the tyrannosaur *Tyrannosaurus rex* [18,19,20], the ovirapotorosaurs *Heyuannia huangi* [21,22] and *Citipati osmolskae* [23], the alvarezsaurid *Shuvuuia deserti* [24], the dromaeosaur *Sinornithosaurus* [17], the early-branching avialan *Anchiornis huxleyi* [25], the early-branching sauropodomorph *Lufengosaurus sp.* [3], an unidentified titanosaurid dinosaur [26], the ankylosaur *Borealopelta markmitchelli* [27], the ceratopsian *Psittacosaurus* [28], the hadrosaur *Brachylophosaurus canadensis* [29,30], an indeterminate hadrosaur material [26], and *Hypacrosaurus stebingeri* [31]. Here, we review the chemistry of the organic molecules recovered to date from fossilized non-avian dinosaurs and discuss the analytical methods used for their detection.

## 2. Analytical Techniques to Investigate Preserved Organic Compounds

The principles of the analytical techniques used in paleontological research, along with their advantages and drawbacks, have recently been reviewed in detail [32,33]. The application of mass spectrometry in proteomic analysis of fossils was specifically discussed by Schweitzer et al. (2019) [34]. The following paragraphs present selected analytical techniques that have been utilized to detect organic compounds in fossilized dinosaurs.

### 2.1. Microscopy

Initial studies carried out on fossils in search of organic matter included a thorough screening of the fossils’ surface or of petrographic thin sections to identify regions in which soft tissues and associated organic compounds could be preserved [35]. Imaging techniques such as optical microscopy (OM), scanning electron microscopy (SEM), and transmission electron microscopy (TEM) have been used for this purpose. Optical microscopy is useful for the visualization of petrographic thin sections to identify preserved cellular structures. Mineralization, diagenetic alteration, and/or microbial contamination of tissues can be detected by means of this technique [35]. 

Electron microscopy is more powerful due to its much higher resolution. It is therefore used to examine subcellular structures in greater detail. In SEM, electrons are directed onto the surface of the sample, generating and transmitting secondary electrons to a detector. Therefore, SEM is limited to studying the sample surfaces by generating a pseudo-3D gray-scale topographical image without collecting chemical signals [35,36]. However, a technique known as energy-dispersive X-ray spectrometry (EDS is often combined with SEM, which uses high energy X-rays characteristic for a specific element released alongside the secondary electrons [32,35,37]. Integration of the elemental information from EDS into the topographical map from SEM allows the localization of elements to be identified in the sample [35]. Other variations of SEM exist, such as field emission SEM (FESEM) [35] and variable pressure SEM (VPSEM) [38]. VPSEM allows for analysis of uncoated samples within a wider range of beam energies than traditional SEM [35]. VPSEM can also be used without prior sample preparation (e.g., dehydration or drying) in soft samples [39]. Thus, FESEM and VPSEM reduce the risk of sample contamination. Both techniques have been used for the study of soft tissues in dinosaur bones [35,38].

In transmission electron microscopy (TEM), electrons are directed to partially demineralized or very thin-cut sections of a sample in a way that only the electrons that cross through the sample are detected. This feature makes TEM a high-resolution technique that can be used for identifying subcellular structures such as organelles or characteristic structural patterns, e.g., the 67 nm bands of collagen fibers [35]. 

### 2.2. Spectroscopy and Spectrometry

#### 2.2.1. UV/Vis Spectroscopy

Ultraviolet/visible light (UV/Vis) spectroscopy is an analytical technique to measure the absorption, transmittance, or reflectance of light by molecules upon irradiation with ultraviolet (190–380 nm) or visible (380–750 nm) light [40,41]. The functional group(s) of the molecule responsible for light absorption is known as the chromophore, e.g., due to conjugated C=C double bonds and/or aromatic rings. The chromophore contains valence electrons having low excitation energy, which become excited and transit to higher energy levels when the molecule is irradiated [41]. The wavelengths at which light is absorbed can be used to identify the structure of a compound. The amount of light absorbed is directly proportional to the concentration of the compound and thus allows for its quantification [42]. UV/Vis spectroscopy is frequently used in molecular paleontology, particularly when analyzing colored fossils, to detect characteristic absorption bands of pigments; it has, for example, been used for detecting heme [18].

#### 2.2.2. Infrared and Raman Spectroscopy

Further studies on fossils use chemical imaging techniques such as Fourier transform infrared spectroscopy (FTIR) and Raman spectroscopy to search for chemical signals, e.g., of functional groups (e.g., amide or carbonyl group) in the samples [9,21,32]. FTIR excites the vibrations of chemical bonds using infrared irradiation. Each type of chemical bond will absorb infrared (IR) waves in a distinct wave number range in the near-IR (12,500–4000 cm^−1^), mid-IR (4000–400 cm^−1^), or far-IR (400–10 cm^−1^) regions. Most of the important chemical signals that are indicative of functional groups will be present in the mid-IR range [43]. FTIR can be combined with light microscopy to identify the location of the detected functional groups in the sample. However, FTIR entails many disadvantages. The wave number ranges can overlap if the sample contains many organic signals leading to frequent misinterpretations of chemical signals, especially if diagenetic changes occurred to the original structure. In addition, any contamination on the surface of the sample will be recorded in the spectra and may not be distinguishable from the sample signals. FTIR has been used to detect characteristic absorption bands of peptide bonds, amide I (C=O bond, ca. 1655 cm^−1^) and amide II (N-H bond, ca. 1545 cm^−1^), associated with collagen in cartilage, in addition to peptide bonds specific for melanin (1580 cm^−1^) [32,34]. 

Other variants of IR spectroscopy have been used to study fossils. For example, synchrotron-radiation Fourier transformed infrared spectroscopy (SR-FTIR) uses a much brighter light source (synchrotron radiation) ranging from far-IR to near-IR [44] to produce chemical maps. SR-FTIR has a higher resolution and a better signal-to-noise ratio than classical FTIR [45]. In addition, attenuated-total reflection IR (ATR-IR) has been used for the analysis of liquid samples [46].

Raman spectroscopy applies monochromatic laser light (ultraviolet, infrared, or visible) to irradiate the layer directly below the surface of the sample. Some of the light is then scattered with a defined frequency generating a signal that can be detected and plotted as a graph of intensity versus wave number. The observed scattering depends on the type of functional group and its vibration [47]. It can be combined with other microscopic techniques such as confocal microscopy to form a chemical map of the functional groups present in the sample. Raman spectroscopy is currently one of the most preferred methods to search for preserved organic matter and other chemical constituents in fossils because it does not require exhaustive sample preparation. However, in contrast to FTIR, signals present on the outermost surface cannot be detected. The produced signals are weak, often requiring prolonged periods of intense irradiation [48], which can lead to a degradation of thermolabile compounds due to the heat produced by the laser [32,49]. Raman spectroscopy has been used for the detection of heme in dinosaur bones [18] and for the detection of the heme degradation product biliverdin and of its precursor protoporphyrin IX in dinosaur eggshells [21].

#### 2.2.3. Mass Spectrometry

Mass spectrometric techniques are among the most sensitive, reliable methods to detect organic compounds. Soft ionization techniques allow measuring the mass-to-charge ratio of intact molecular ions. In addition, different chemical classes of compounds have characteristic fragmentation patterns observed in mass spectrometry [50]. However, detecting only fragments or only molecular ions is often not sufficient for identification of specific organic molecules [32,35], whereas a combination of both can be highly informative. 

Chemical information, especially on molecular fragments, can be obtained by time-of-flight secondary ion mass spectrometry (TOF-SIMS) and pyrolysis coupled to gas chromatography-mass spectrometry (Py-GC-MS). Only fragments can be detected by the latter method because of the harsh ionization conditions used, often leading to a complete destruction of the sample.

TOF-SIMS is a surface imaging technique with ultra-high spatial resolution which directs high energy ionizing beams (e.g., gallium ions) over the sample surface. Molecules are released, ionized and often fragmented [51]. The ions are transmitted to the time-of-flight mass spectrometer and detected according to the time it takes for them to reach the detector. The heavier their masses are, the more time it will take. It can be used for analyzing fragile or small amounts of fossil samples because measurements take place at the surface without the need for extractions. Determination of the location of the signal in the sample is the main advantage of the method, and it is therefore useful for organic compound screening [52]. However, as TOF-SIMS only analyzes the surface, any changes on the surface or contamination will influence the results [32,35]. This method has been used to detect heme [53], melanin [54], protein fragments of β-keratin [24], and collagen [26] in fossils.

To overcome the extensive fragmentation, especially of higher molecular weight ions, a variant of TOF-SIMS known as cluster secondary ion mass spectrometry was developed. Its principle relies on bombardment of the sample using a polyatomic cluster of ions, such as gold (Au_3_) or a C_60_-based ion cluster, buckminsterfullerene. This allows the detection of intact molecular ions in the range of 1000–3000 D, which was not possible with traditional TOF-SIMS [55]. In addition, spatial resolution beyond the micrometer range can be achieved [56]. 

Py-GC-MS is a technique that uses intense heat (ca. 400–600 °C) to fragment molecular bonds. The generated fragments are gaseous; they are separated by gas chromatography and detected by mass spectrometry. Unlike TOF-SIMS, the sample is destroyed, and the location of the chemical signal in the original sample cannot be determined. This method does not require sample preparation, and therefore, the risk of detecting artifacts is lowered [57]. It has been used to detect molecular fragments characteristic of proteins, lignin and chitin in fossils [58,59,60]. Due to the destruction of the sample, Py-GC-MS is not preferred if alternative approaches are possible; therefore, it is only used for analyses of insoluble fossil material which cannot be analyzed otherwise [35].

All of the aforementioned analytical techniques are often not suitable for unambiguously determining the identity of organic compounds. However, they can help narrowing down sample regions that contain organic compounds, which may then be subjected to more invasive mass spectrometry techniques. 

On rare or unique fossils, only non-destructive or highly sensitive methods can be applied. Modern mass spectrometry techniques now provide options for analyzing such precious samples since they require only small quantities of material. 

In order to identify intact organic compounds, mild ionization methods, such as electrospray ionization, need to be applied [61]. In most cases, samples are extracted, separated by reverse-phase liquid chromatography, which is coupled to tandem mass spectrometry (LC-MS/MS). The prerequisite for this type of analysis is a solution of the analytes; thus, compounds that are insoluble in the typically used solvents (methanol, acetonitrile, water, and their mixtures) cannot be analyzed [62].

The type of mass analyzer used is decisive for mass accuracy and sensitivity of mass spectrometric measurements. Quadrupole, time-of-flight, linear ion-trap, Fourier transform ion cyclotron resonance (FT-ICR) and Orbitrap analyzers are commonly used for organic compounds in fossils. Instruments with high mass accuracy are needed to determine elemental compositions of organic compounds. With ion trap instruments or when two types of mass analyzers are combined in series, more advanced mass spectrometric analyses are possible, known as tandem mass spectrometry. Typical combinations are quadrupole/quadrupole, quadrupole/time-of-flight (q/TOF), and quadrupole or linear ion-trap coupled to Orbitrap. Tandem mass spectrometers allow for a unique type of analysis known as collision-induced dissociation, in which intact ions of a defined mass-to-charge ratio are selected and then deliberately fragmented to analyze the fragments [62]. This method is used to achieve ultra-high sensitivity, and it provides structural information on the molecules of interest. It is the method of choice in proteomics to identify peptide sequences and to obtain information about diagenetic changes to the chemical structure, and to identify post-translational modifications [34,35]. For fossils, LC-MS/MS is one of the most selective, accurate and sensitive methods to identify organic compounds. However, this is often not applicable due to limited sample availability and/or difficulties in extracting the target compounds due to a lack of solubility [33,35]. 

### 2.3. Immunological Techniques

Immunological techniques are based on antigen–antibody reactions. Antibodies used in the process are specific to a certain epitope in the target tissue. These sensitive techniques are used to screen for the presence of macromolecules such as proteins or DNA. Using antibodies, sequence determination is not possible, but regions in the sample may be located, in which proteinaceous or genetic material has been preserved, and which can be selected subsequently for mass spectrometric analysis [34]. Immunological techniques include enzyme-linked immunosorbent assays (ELISAs), Western blotting (immunoblotting), and immunohistochemistry/immunostaining procedures. 

A prerequisite for the detection of proteins by Western blot and ELISA is a liquid extract containing the protein of interest. ELISA is the more sensitive technique [63]. There are different forms of ELISA: direct ELISA, indirect ELISA, sandwich ELISA and competitive ELISA, which are typically performed in well plates. The first step is to immobilize the antigen of interest by direct adsorption to the surface or through binding to a capture antibody fixed to the plate. Direct and indirect ELISA are used for antigens immobilized directly to the well plate, whereas sandwich ELISA is used for antigens bound to a capture antibody [64]. Direct ELISA uses an enzyme-linked antibody that binds directly to the antigen of interest. Upon washing to remove unbound antibodies, and subsequent addition of the suitable substrate, a color change will occur only in the wells that contain the antigen–antibody complex [65]. Indirect ELISA is used to detect the presence of antibodies rather than antigens. Addition of a sample expected to contain a primary antibody specific to the antigen of interest results in the formation of a complex with the immobilized antigen. A secondary antibody linked to an enzyme and specific to the primary antibody is added. After washing, any unbound antibodies are removed. The substrate is added and the enzymatic reaction occurs to produce a colored product that confirms the presence of the antibody [64,65]. 

Sandwich ELISA is used to detect the presence of antigens and is the most commonly used form of ELISA. The well surface is first coated with a capture antibody specific to the antigen of interest, onto which the antigen from a sample will be immobilized. A primary antibody specific to the antigen will then be added. If the antigen is present, the primary antibody will bind to it. The next steps are the same as those for indirect ELISA, by which the color change will confirm the presence of the antigen [64,65]. It is worth noting that sandwich ELISA will only be possible for antigens which have two separate epitopes for binding a capture antibody and a primary antibody. Using two antibodies for detection of the same antigen makes sandwich ELISA highly specific [66]. In ELISA, proteins are detected in their natural conformation.

In competitive ELISA, antigens in a sample compete with a reference antigen coated on the surface of a well in binding to a labeled primary antibody of known concentration. The sample is incubated first with the primary antibody. Then this solution is added to the wells. The more antigens are present in the sample, the more primary antibodies will bind to them [67]. Any unbound antibody will then bind to the reference antigen. Following a washing step, an enzyme-linked secondary antibody is added. The substrate for the enzyme is then added, and the intensity of the resulting color is inversely related to the concentration of the antigens present in the sample. If few primary antibodies are bound to the reference antigen, a faint color will be observed, and this indicates a high concentration of antigens in the sample [68].

Western blot is used to identify a protein from a complex mixture [64]. Before performing a Western blot experiment, the mixture of proteins in a sample are separated by polyacrylamide gel electrophoresis according to size [69,70]. There are two main types of gel electrophoresis, depending on the type of additives used: sodium dodecyl sulfate-polyacrylamide gel electrophoresis (SDS-PAGE) and blue native-polyacrylamide gel electrophoresis (BN-PAGE). SDS-PAGE uses the detergent sodium dodecyl sulfate which denatures the proteins, whereas BN-PAGE uses the mild Coomassie blue dye and does not denature the protein of interest [69,71,72]. The bands containing the separated proteins are transferred to an immobilizing nitrocellulose or polyvinylidene difluoride membrane. This is followed by adding a blocking buffer containing non-fat dried milk or 5% bovine serum albumin, in order to prevent binding of antibodies to the membrane [70]. A primary antibody specific to the protein of interest is incubated with the membrane, followed by washing to remove unbound antibodies. Then, a secondary antibody is added that binds specifically to the primary antibody, and which is radiolabeled or linked to an enzyme. Afterwards, either a substrate is added to initiate the enzymatic reaction, or a photographic film for a radio-labeled substrate is used for detection of the target antigen–antibody complex and to locate the protein [64]. 

To account for diagenetic changes to the original structure, polyclonal antibodies are often used during analysis of fossil extracts; however, problems with poor specificity of antibodies may arise. Both Western blot and ELISA are prone to contamination and/or interference from extraction buffer components [34]. 

Immunohistochemistry is based on the same principles as ELISA and Western blot, the only difference is that the antibodies are applied in situ on intact tissue instead of utilizing extracts [73]. In situ analyses are preferred to destructive techniques because they minimize the loss of precious sample material and/or degradation of organic material during preparation (e.g., after exposure to chemicals or air) [34]. Suitable microscopic tissue slides containing the epitopes of interest are fixed, usually by formalin, into a polymer or paraffin wax [74]. If the fixation process is known to mask the antigens of interest, an extra step is usually performed by physical (e.g., heat or ultrasound) or chemical (e.g., enzymatic digestion) methods to break any cross-links formed, making the antigens re-accessible to antibodies [75]. The next step is incubation with a blocking buffer such as bovine serum albumin to prevent non-specific binding. This is followed by adding primary antibodies specific to the antigen of interest, then washing to remove unbound antibodies. Fluorescence-labeled or enzyme-linked (e.g., peroxidase or alkaline phosphatase) secondary antibodies are then added [73]. Visualization of positive reactivity takes place by light or fluorescence microscopy, or after addition of substrate and monitoring of the color change due to the enzymatic reaction. This immunological assay allows for the localization of target antigens in tissues, which is not possible with ELISA and Western blot techniques [74]. 

## 3. Organic Compounds Found in Dinosaurs

The following sections will describe the evidence and chemistry of organic compounds found to date in non-avian dinosaurs. An overview of the localities and age of the dinosaurs is depicted in Figure 1.

### 3.1. Pigments

Pigments are molecules that absorb light of wavelengths in the visible range (ca. 380–750 nm) and, accordingly, are responsible for the colors seen in many organisms and some minerals. Examples of naturally occurring pigments or biochromes are porphyrins, melanins and carotenoids [77]. Recent research has focused on investigating the preservation of pigments that are responsible for colors seen in fossils. Based on molecular analyses, scientists have been able to reconstruct the original color of some dinosaurs, also referred to as paleocolor reconstructions [78,79]. The pigments believed to have been preserved in dinosaur fossils include porphyrins (heme and protoporphyrin IX), their open-chain tetrapyrrole derivatives (biliverdin) and the biopolymer melanin (eumelanin and pheomelanin).

#### 3.1.1. Porphyrins

Porphyrins are a family of organic compounds containing four pyrrole rings connected by methine bridges. Examples are heme (**1**), the iron-complexing main prosthetic group of hemoglobin, and protoporphyrin IX (**2**), the metal-free precursor of heme (see Table 1 for structures). Metabolic degradation products include linear tetrapyrrole derivatives, e.g., biliverdin (**3**) (see Table 1). Porphyrins and their derivatives are relatively stable, even for hundreds of millions of years; they have been recovered from sediments and crude oil extracts, the oldest record being from 1.1-billion-year-old sediments [80]. Porphyrins have also been detected in fossil tissues from dinosaurs [eggshells [22] and trabecular bone [18]] and the abdomen of a female mosquito [53]. The chemistry of porphyrins in fossils has been recently reviewed [81]; the porphyrins detected in fossils derived from dinosaurs are compiled in Table 1.

Heme was identified in trabecular bone extracts of *Tyrannosaurus rex* in 1997 [18]. The distinct chemical feature which made it possible to confirm the identity of heme was its chromophore in the ultraviolet/visible light range [18]. The porphyrin ring has a very characteristic band in the ultraviolet range of around 410 nm, known as the *Soret* band, which could be detected using ultraviolet/visible light (UV/Vis) spectroscopy. This band was observed in bone extracts but not in controls, indicating that the signals were derived solely from the bone and not from contaminating factors in the surrounding sandstone sediment or extraction buffers. In addition, four of the six characteristic Raman peaks for hemoglobin (marker bands I, II, IV, and V) were detected with high intensity in the extracts. The six marker bands are found in the following spectral regions: band I (1340–1390 cm^−1^), band II (1470–1505 cm^−1^), band III (1535–1575 cm^−1^), band IV (1550–1590 cm^−1^), band V (1605–1645 cm^−1^), and band VI (1560–1600 cm^−1^) [82]. Resonance Raman spectroscopy analyses on extracts also showed that iron was present in the oxidized ferric state, which indicates a diagenetic alteration of heme (Fe^2+^ complex) to the oxidized hemin form. In addition, proton NMR spectra on the fossil extract were similar to those from degraded hemoproteins containing ferric iron [18].

A further case of heme in the fossil record, although not in dinosaurs, was reported 16 years later, when traces of heme were found in the abdomen of a female fossil mosquito (46 Ma), analyzed in situ by TOF-SIMS [53].

Only recently, the metal-free porphyrin, protoporphyrin IX (**2**), and the linear tetrapyrrole derivative biliverdin (**3**) were detected in extracts of eggshells from the oviraptorid dinosaur *Heyuannia huangi* by liquid-chromatography electrospray ionization-quadrupole-time-of-flight mass spectrometry (LC-q/TOF-MS) [22]. The exact masses were detected with high resolution in the mass spectra as protonated molecular ions, [M + H] ^+^, from three fossil eggshell samples. For confirmation, extant emu eggshell extracts and commercial standards of the two compounds were also analyzed. These peaks were not detected either in the sediment samples or in control samples, indicating that the peaks truly belonged to the analyzed fossil. Protoporphyrin IX (**2**) is more hydrophobic than biliverdin (**3**) and therefore more likely to be preserved due to its resistance to hydrolytic attack. In addition, the ring system of protoporphyrin is more stable than the open chain structure of biliverdin. Based on these results, a reconstruction of eggshell color as blue-green was performed [22]. A year later, protoporphyrin IX and biliverdin were reported using Raman spectroscopy in various fossilized eggshells, including *Heyuannia huangi* [21]. This study has been criticized by experts in Raman spectroscopy because the authors had based their observations only on a single analytical technique; Alleon et al. even argued that the observed signals were due to instrumental artefacts caused by background luminescence, and not due to Raman scattering [83,84].

There appears to be still much potential for future discoveries of porphyrins and their metabolites and degradation products in dinosaurs and other fossils.

#### 3.1.2. Melanins

Melanins are a group of dark-colored biopolymeric structures. Different types of melanin are known: eumelanin (**4**), pheomelanin (**5**), allomelanin, pyomelanin and neuromelanin (see Figure 2). Eumelanin, pheomelanin, and allomelanin are most relevant when studying fossils. Eumelanin (**4**) and pheomelanin (**5**) are nitrogen-containing melanins found in animals. Allomelanin is a nitrogen-free melanin (see Figure 3) which is found in plants, fungi and bacteria; it is relevant when studying fossils because detection of its chemical signals can imply external microbial contamination [25].

The biosynthesis of eumelanin and pheomelanin takes place in melanocytes in the dermis. Figure 2 shows the biosynthesis of eumelanin and pheomelanin, including their intermediates. The synthesized melanins are transported into the keratinocytes, found in the epidermis, in special lysosome-like vesicles known as eumelanosomes and pheomelanosomes [85,87]. Both types of melanosomes are then incorporated into the outer layer of the skin, determining the color of skin, hair, and eyes. Melanins are responsible for absorbing UV light and for scavenging free radicals that can be formed upon exposure to UV light, in order to protect the inner layers of the skin from harmful radiation and radical reactions [86,88].

Eumelanin is brown to black in color and contains repeating units of 5,6-dihydroxyindole (**6**) and 5,6-dihydroxyindole-2-carboxylic acid (**7**). In its biosynthesis, it is derived from the amino acid tyrosine (**8**), which, upon action of tyrosinase, or by oxidation, is converted to DOPA-quinone (**9**), which is then cyclized and decarboxylated to form 5,6-dihydroxyindole (**6**) through the intermediate compounds leucodopachrome (**10**) and dopachrome (**11**) [25,85,86] (for structures see Figure 2). Some indole units may randomly undergo partial oxidative cleavage via formation of an ortho-benzoquinone leading to pyrrole-di-carboxylic acid derivatives, which are incorporated into the polymeric structure of eumelanin [89,90]. Pheomelanin is a reddish-yellow sulfur-containing melanin which contains units of 1,4-benzothiazine and 1,3-benzothiazole [91]. Similar to eumelanin, pheomelanin is derived from tyrosine (**8**), and additionally from cysteine (**12**), that is fused with DOPA-quinone (**9**) to form cysteinyl-DOPA derivatives **13** and **14**, which undergo several oxidation steps to form 1,4-benzothiazine intermediates **15** and **16** [85] (see Figure 2). 

Allomelanin has not been studied as much as eumelanin and pheomelanin. However, it is established that several subtypes of allomelanin can be distinguished according to the precursors from which they are derived. The precursors comprise 1,8-dihydroxynapthalene (**17)**, 1,4,6,7,9,12-hexahydroxyperylene-3,10-quinone (**18**) and biphenolic dimers such as 3,3′,4,4′-tetrahydroxy-1,1′-biphenyl (**19**), biosynthesized from acetyl-CoA, malonyl-CoA, and catechol, respectively (see Figure 3). Accordingly, three types of allomelanin are distinguished:1,8-dihydroxynapthalene melanin, 1,4,6,7,9,12-hexahydroxyperylene-3,10-quinonemelanin, and catechol-melanin [92].

There is emerging morphological and chemical evidence for eumelanin and pheomelanin detected in a variety of fossils with or without association with melanosomes. Examples are fossilized marine reptiles such as a Paleogene turtle (55 Ma), Cretaceous mosasaur (86 Ma), and Jurassic ichthyosaur (ca. 196–190 Ma) [93]. The compounds were also found in several species of fish (359–366 Ma), amphibians (Ypresian/Lutetian, Eocene, Aquitanian, Miocene, Chattian, Oligocene), birds (56–34 Ma), and mammals (56-34 Ma) [94]. Furthermore, they were detected in dinosaurs (150–112 Ma) [25,27]. A summary of findings on melanins and/or melanosomes in the dinosaur fossil record is compiled in Table 2, along with the analytical methods used. 

Imaging studies using SEM in combination with EDS have been used to detect melanin based on the presence and shape of melanosomes in preserved integumentary structures of the theropods *Sinosauropteryx* and *Sinornithosaurus* [17], as well as *Psittacosaurus* [28]. More recently, analytical techniques such as TOF-SIMS and Py-GC-MS have been utilized to confirm the chemical fingerprint of melanin in the early avialan *Anchiornis huxleyi* [25] and the ankylosaur *Borealopelta markmitchelli* [27]. Due to the resemblance between melanosomes of dinosaurs and keratinophilic bacteria on the microscopic level [17,95], a chemical analysis is necessary in order to confirm the presence of melanin [25]. 

TOF-SIMS analyses of a feather fossil derived from *Anchiornis huxleyi* (150 Ma) showed negative ion spectra characteristic for melanins in the areas where microscopic melanosome-like structures were observed [25]. Compared to spectra of synthetic and natural variants of eumelanin and pheomelanin, many high-intensity mass signals were in common, indicating the presence of eumelanin of animal origin. Absorption bands suggesting the presence of eumelanin as well were detected using infrared spectroscopy. Bacterial contamination was excluded in the examined areas due to the absence of peaks corresponding to peptidoglycans and hopanoids [25]. Peptidoglycans, polymers consisting of sugars and peptides, are cell wall components of Gram-positive and Gram-negative bacteria [96], while hopanoids are cyclic lipophilic triterpenoids that are located in the bacterial cell membrane and have been detected in the fossil record of bacteria [97,98]. The TOF-SIMS spectra of bacteria-derived melanin, namely allomelanin, which does not contain nitrogen (see Figure 3), does not show any of the nitrogen-derived peaks that were found in the fossil (mass-to-charge ratios of 50, 66, 74, 98, 122, and 146). Analysis of the surrounding sediment using the same method showed negative ion spectra corresponding to silicate-rich minerals, but no nitrogen-containing peaks were observed. Signals for sulfur-containing compounds that could originate from pheomelanin were not intense enough to confirm its presence in the fossil [25].

The preserved integumentary structures of the ankylosaur *Borealopelta markmitchelli* (112 Ma) were analyzed by TOF-SIMS and pyrolysis-GC-MS to investigate the presence of melanin [27]. TOF-SIMS analysis showed negative ions similar to those of melanin in previously reported fossils [93], resembling natural and synthetic melanin. In addition, ions containing sulfur (1,3-benzothiazole) indicative of pheomelanin [93] were detected, suggesting that a mixture of eumelanin and pheomelanin was present [27]. Pyrolysis-GC-MS analysis showed signals corresponding to eumelanin (N- and O-heterocyclic and aromatic compounds), as reported previously in fossils [99,100]. Signals derived from pheomelanin (1,3-benzothiazole) were also present, which were not detected in the surrounding sediment [27].

### 3.2. Proteins

Although met with controversy, especially when considering chemical instability, there are more and more reports on proteins and their fragments detected in fossils. In the early years, this was backed mainly by morphological examination and the application of vibrational spectroscopy and immunological techniques. In recent years, the field of paleoproteomics has flourished, applying high-resolution mass spectrometry to determine peptide sequences and to map them on the extant versions of the proteins of interest [33,34,101]. Further paleoproteomic research, especially sequencing of proteins by mass spectrometry, would be required to confirm the endogeneity of the detected protein fragments [34]. It has to be kept in mind that cross-contamination remains an important issue when analyzing peptide sequences [102]. Not only can cross-contamination arise from laboratory reagents and controls, it can also occur due to previously analyzed samples. Thus, it is necessary to rule out cross-contamination by suitable measures, such as careful and self-critical approaches, and appropriate controls [102].

Vibrational methods such as infrared spectroscopy have been used to detect proteins, showing characteristic absorption bands of the amide bonds; however, these signals are non-specific and it is not possible to identify the type of protein or its sequence [34]. Early trials to detect proteins utilized amino acid analysis after degradation of the proteins. This method is also insufficient for determining the original peptide sequence [33]. TOF-SIMS employs a harsh ionization method which causes extensive fragmentation of proteins. Therefore, while it cannot be used for sequencing, it is useful for obtaining a chemical map, revealing the regions where amino acid fragments are found in a fossil, which may then be further analyzed [32,34,76]. 

Immunological techniques including immunohistochemistry, Western blotting, and ELISA rely on positive antigen–antibody reactions, detecting specific epitopes of a protein or nucleic acid. Specificity depends on the employed antibodies, but protein or nucleic acid sequences cannot be determined. These methods can be useful to locate the regions that may contain preserved proteins (or nucleic acids) suitable for subsequent mass spectrometric analysis. In addition, liquid chromatography coupled to electrospray ionization high-resolution mass spectrometry is used for the identification of proteins. The techniques used in paleoproteomics and their limitations were recently reviewed [33,34]. Most of the proteins detected in dinosaur fossils belonged to the most abundant ones including collagen type I (found in bones), collagen type II (found in cartilage), and beta-keratin (found in scales, turtle shell, claws of reptiles, and in avian feathers) [103]. The following section will discuss the evidence for proteins detected to date in dinosaurs and their chemistry. 

#### 3.2.1. Collagens

Collagens constitute a family of glycoproteins that are the main components of the extracellular matrix of different tissues. In animals, 29 different types of collagen have been found, but only 3 types (collagen I, II and III) constitute around 80–90% of the total collagen. Collagens are structural proteins in the extracellular matrix which confer mechanical strength especially to connective tissues. They directly interact with other components of the extracellular matrix, such as proteoglycans, fibronectin and laminin. Proteoglycans are glycoproteins that form a gel-like network in the extracellular matrix. Collagens and other fibrous proteins (fibronectin and laminin) are located within this network. Fibronectin and laminin are non-collagenous glycoproteins that form fibrous networks and affect the shape of the extracellular matrix. They possess binding sites important for cell adhesion [104]. In addition, collagens interact with secreted soluble factors such as the von Willebrand factor and interleukin-2, and with cell surface receptors such as integrins [105]. These interactions aim to regulate tissue development and mechanical responses to cell signaling such as cell adhesion, migration and chemotaxis [106,107]. The primary polypeptide structure of collagen is known as the α-chain. All types of collagens share the repeating amino acid sequence [Gly-X-Y], where X and Y are usually proline and hydroxyproline, respectively (see Figure 4). In 12% of collagen sequences, both proline and hydroxyproline are present in their respective positions, while in 44% of the sequences, only one of them is present [108]. The secondary structure of collagen is formed from three α-chains arranged in parallel. They are twisted together to form the tertiary structure, a rope-like triple helix with a molecular weight of ca. 300 kDa, a length of 280 nm and a diameter of 1.4 nm. The abundance of the cyclic amino acids, proline and hydroxyproline, sterically hinders rotation around the peptide bonds in the α-chains which contributes to the stability and rigidity of the triple helix. In addition, two types of hydrogen bonds stabilize the triple helix. The first type of intermolecular hydrogen bonds are formed between the NH of glycine and the carbonyl group of proline residues in neighboring α-chains. The second type are intramolecular hydrogen bonds, formed between the carbonyl or hydroxyl groups of hydroxyproline and the carbonyl group of glycine or hydroxyproline residues in the same α-chain, mediated by a water molecule [108]. Moreover, the X and Y positions in further collagen sequences are occupied by other amino acids, but never contain tryptophan, tyrosine, or cysteine, as these would destabilize the triple helix [109]. Post-translational modifications such as hydroxylation of proline and lysine residues or glycosylation (with galactose or a disaccharide of glucose and galactose) also contribute to stability and are typical of collagen. The more hydroxyproline residues there are, the more thermally stable the triple helix is. Post-translational hydroxylation of proline residues is often used in identification of collagen from fossils, especially since this cannot be performed by bacteria [108]. Hydroxylysine is the point of attachment of the sugars via an O-glycosidic linkage, and this stabilizes the collagen fibrils mechanically by formation of covalent crosslinks [108]. 

##### Collagen Type I

Collagen type I is the major component of bone organic phase, but is also present in skin, tendons, ligaments, lung, blood vessels, cornea, brain and spinal cord [108,112] (see Figure 4 for structures). Collagen I is composed of two α1(I) chains and one α2(I) chain and assembles into elongated fibrils of 500 µm in length and 500 nm in diameter. The fibrils have a characteristic tight arrangement. Every 64–67 nm, there also is a pattern repeating itself, known as *D*-banding. This pattern is visible in the electron microscope and can be utilized for identification of collagen type I [108,113,114,115,116]. Studies reporting evidence for collagen type I found in the dinosaur fossil record are compiled in Table 3.

Attempts to detect collagen type I in dinosaurs were performed on samples of *Tyrannosaurus rex* [19] (see Table 3). Trabecular bone extracts showed positive reactivity in an ELISA employing avian collagen I antibodies. The signal was weaker in the dinosaur as compared to extant emu cortical and trabecular bone, but the signal detected in the fossil was larger than those in buffer controls and in the sediment. The same pattern was observed by in situ immunohistochemistry studies. Antibody binding decreased significantly when the fossil tissue was digested with collagenase I before exposure to the antibodies. TOF-SIMS analysis revealed amino acid residues in the fossil including glycine (highest relative signal intensity), alanine, proline, lysine, leucine and isoleucine [19]. A subsequent study [20] applied a softer mass spectrometric technique to avoid undesired fragmentation, liquid chromatography tandem mass spectrometry (LC-MS/MS). In this study, the dinosaur fossil was compared to similarly treated ostrich and mastodon samples. The mass spectra obtained from *T. rex* bone extracts detected seven collagen peptide sequences, five from the α1(I) chain, one from the α2(I) chain, and one belonging to the α1(II) chain of type II collagen, that were aligned with database sequences from extant vertebrates. Post-translational modifications, especially hydroxylation of proline, lysine and glycine, were detected in the dinosaur fossil as well as in the mastodon and ostrich samples, while no collagen sequences were detected in control samples of the surrounding sediment and the extraction buffers. The sediment contained peptides of bacterial origin, but no collagen [20]. 

In another study, investigation of the hadrosaurid dinosaur *Brachylophosaurus canadensis* provided evidence for collagen type I [29]. This was confirmed by studies in different laboratories and at different times using different methodology including sample preparation technique, mass spectrometry instrument, and data analysis software [29,30]. Microscopic observation (by field-emission SEM) of fibrous structures in demineralized femur bones was followed up by immunoblot assays. A positive reactivity to antibodies raised against avian collagen type I was observed in whole fossil bone extracts and in intact demineralized fossil bones [29]. In situ immunohistochemistry studies performed on demineralized fossil bones confirmed the results. The extraction buffers and the surrounding sediments showed no reactivity. Antibody binding decreased significantly when the samples were digested with collagenase before exposure to the antibodies, or when exposed to antibodies that had been pre-incubated with excess collagen. Gel electrophoresis studies on samples of the surrounding sediment did not show any visible protein bands. Infrared spectroscopy showed absorption bands of amide bonds (Amide I and Amide II). Analysis using TOF-SIMS indicated fragments of lysine, proline, alanine, glycine, and leucine residues in intact blood vessels and in matrix of demineralized bone. Further experiments using reversed-phase microcapillary liquid chromatography tandem mass spectrometry (linear ion-trap alone or hybridized with Orbitrap mass spectrometry) recovered eight collagen type I sequences, containing a total of 149 amino acids. Six of these sequences were attributed to the α1 chain and two to the α2 chain [29]. 

High-resolution measurements performed eight years later by LC-tandem mass spectrometry coupled to Fourier-transform ion cyclotron resonance mass spectrometry (FT-ICR-MS) again showed eight collagen type I sequences in the range of 250 kDa, two of which had previously been detected for the α1 chain, in addition to three new α1 chain sequences, and three new sequences for the α2 chain [30]. In both studies, no collagen sequences could be detected in spectra of extraction buffer or samples of the surrounding sediment. In addition, post-translational modification of hydroxylated proline was observed, which is important for the triple helix structure of collagen I and cannot be produced by microbes [29,30]. 

Amino acid fragments in association with direct observations of fibrous structures showing the 67 nm banding of typical collagen were detected using TOF-SIMS by Bertazzo et al. (2015) in a variety of dinosaur bone samples from the Late Cretaceous Dinosaur Park Formation of Alberta, Canada [26]. The banding indicated that the quaternary structure of collagen may have been preserved. In addition, TOF-SIMS analyses were performed to search for amino acids using thick sections of the fossil bone sample, as well as modern rabbit bone, non-calcified fossil samples, surrounding sediment, and the sample holder made of copper as controls. The fossil dinosaur bone samples and the rabbit bone contained similar amino acid peaks which were neither present in the non-calcified fossil samples nor in the surrounding sediment or in the sample holder. Fragments belonging to glycine, arginine, alanine, and proline were detected only in the permineralized fossil samples [26]. 

Synchrotron-radiation Fourier transformed infrared spectroscopy (SR-FTIR) and confocal Raman spectroscopy were used to identify characteristic vibrations of chemical bonds at specific absorption bands for each functional group, producing high resolution images and spectra [34]. Infrared absorption bands characteristic for collagen type I were detected in thin sections of the rib bone of a 195-million-year-old *Lufengosaurus*, and early-branching sauropodomorph, and the geologically oldest dinosaur sample analyzed to date. The detection was especially in the regions where vascular canals could microscopically be observed. The infrared absorption bands of the fossil samples were very similar to the reference samples of extant collagen I extracted from calf skin [3].

The published evidence for collagen type I and other proteins and their sequences in dinosaurs should still be treated with caution. For example, TOF-SIMS is not suitable for sequencing but can only help to locate samples for subsequent tandem-mass spectrometry experiments. A combination of different analytical techniques is usually needed, combined with the proper controls. Tandem mass spectrometry is the main technique to prove the presence of peptides and to sequence polypeptides/proteins. 

##### Collagen Type II

Collagen type II is a structural protein mostly present in cartilage, tendons, and in the intervertebral disc [108]. In contrast to collagen type I, it is a homotrimer, composed of 3 α1(II) chains [113]. Similar to collagen type I, it also forms a triple helix of around 1000 amino acids in length and has the repeating amino acid pattern of Gly-X-Y [117], forming an aggregated fibrous structure. 

The first report on collagen II associated with preserved calcified cartilage in dinosaurs [31] was from *Hypacrosaurus stebingeri*, a 75-million-year-old hadrosaur nestling discovered in the Two Medicine Formation of northern Montana, USA. Techniques used to chemically characterize the observed chondrocyte-like microstructures were histochemical and immunological techniques, as shown in Table 3. Thin sections of demineralized fossil cartilage exposed to antibodies raised against avian collagen type II showed positive reactivity after visualization by green fluorescence. The observed pattern was interrupted and less intense compared to the homogenous distribution of the binding pattern in extant cartilage from emu (*Dromaius novaehollandiae*), suggesting that either the epitopes are few or that the epitopes recognized by avian collagen I antibodies are not similar to those present in the dinosaur. Collagen II is not produced by bacteria; thus, contamination is less likely to have occurred [31]. 

Specificity of the antibodies was checked by prior digestion of the thin sections by collagenase II and exposure to the antibodies, after which the binding decreased significantly in both fossil and recent material under the same conditions. This supports the interpretation that collagen II is likely present in the fossil. Antibodies against avian collagen I did not show any binding in both fossil and recent cartilage, which is not expected to be found there [31].

#### 3.2.2. Keratins

Keratins are structural proteins which are the major constituents of hair, nails, feathers, horns, and hooves [118]. They are characterized by a high cysteine content (7–13%). Keratins have several biological functions, including (i) mechanical effects and (ii) altering cellular metabolism. By disassembly and reassembly, keratins provide flexibility to the cytoskeletal structure, making cells and tissues withstand mechanical stress and maintain their shape. Keratins affect the response to cellular signaling by binding to various signaling proteins such as protein kinases and phosphatases. Thus, keratins are involved in the regulation of cell growth, cell differentiation, mitosis, and protein synthesis, which may lead to a change in cellular metabolism [119,120,121].

Keratins have a molecular weight of 40–70 kDa [119,122]. The amino acids in the primary sequence of keratins are often cysteine, glycine, proline, and serine, and to a lesser extent lysine, histidine, and methionine. Tryptophan is rarely present [118,123]. The secondary structure of keratins is either an α-helix or a β-sheet, depending on the type of amino acids present. Accordingly, two types of keratins can be distinguished: α-keratin and β-keratin [119] (see Figure 5).

The tertiary structure of keratins is composed of a dimer that forms the building block of keratin filaments. It is stabilized by inter- and intra-molecular interactions such as disulfide bridges, hydrogen bonds, hydrophobic interactions, and ionic bonds [118]. Their quaternary structure consists of self-assembling intermediate filaments having a characteristic electron-lucent region of 7–8 nm in diameter observed under the electron microscope. The formation of keratin filaments is affected by pH and osmolarity [119].

Post-translational modifications occur to the secondary structure of keratins, which in turn affect their overall structure, physicochemical properties and functions. Phosphorylation or formation of intra- and interchain covalent bonds (e.g., disulfide bonds) can directly modify the structure. Changes in pH, the types of ions present, and osmolarity can alter the physicochemical properties indirectly, for example, by changing the isoelectric point. Keratins can modify their filaments due to mechanical stress such as tension, compression, and shearing [119].

Keratins are insoluble in water, alkali, weak acids, and organic solvents. They are stable in the presence of proteases such as pepsin and trypsin. The crosslinking via disulfide bonds stabilizes the overall tertiary structure and lowers the water solubility [118,123]. 

α-Keratin is expressed in all vertebrates [23]. Its structure is better described than that of β-keratin [119]. α-Keratins are classified into two types according to their isoelectric point (pI) range: type I (pI = 4.9–5.4) and type II (pI = 6.5–8.5). α-Keratins with more acidic amino acids are of type I, while those containing more basic amino acids belong to type II [119] (see Figure 5). 

β-Keratins are exclusively expressed in reptiles and birds (e.g., claw sheaths and feathers), and differ from the α-keratins in their lower solubility and the high rigidity of their microfibril filaments [24]. β-keratin has a core of 30 amino acids and forms antiparallel β-sheets, joined by regions of β-turns and stabilized by hydrogen bonds. The quaternary structure of β-keratin is characterized by microfibril filaments of 3 nm in diameter [125]. The presence of hydrophobic amino acids in the core, such as valine and proline [125], increases their preservation potential because they will not be readily hydrolyzed [23]. β-Keratins are not expressed in humans or microorganisms; thus, if β-keratins are detected in fossils, exogenous contamination can likely be ruled out [23]. β-Keratin has been detected in fossil dinosaurs mainly by immunohistochemistry techniques as shown in Table 4.

There are some amino acids which are common in the sequence of both types of keratins, such as glycine, serine, valine, leucine, glutamate, cysteine, and alanine [119,126]. Amino acids which are more abundant in α-keratin are methionine, histidine, phenylalanine, and isoleucine [127]. Amino acids that are more abundant in α-keratin are proline and aspartate [119], whereas histidine, methionine, tryptophan, and tyrosine are rarely present [126].

The first characterization of β-keratin in fossil dinosaurs was from feather-like structures of the 100-million-year-old *Shuvuuia deserti* collected at Ukhaa Tolgod in southwestern Mongolia (see Table 4) [24]. Immunohistochemical studies using antibodies raised against avian α- and β-keratins showed a strong reactivity in both fossil and extant (duck feather) tissue samples for β-keratin, and less reactivity for α-keratin. No reactivity was seen in control samples, including incubation with antibodies not specific to β-keratin. Reduced binding was observed when the antibodies against β-keratin were incubated with excess β-keratin before exposure to the tissues, thus confirming the specificity of this approach. Furthermore, TOF-SIMS analysis was performed on isolated fiber structures to search for amino acids to support the immunological findings. Several amino acid fragments, containing glycine, serine, leucine, cysteine, proline, valine and alanine, were detected in the mass spectra. The targeted sampling location supports that these amino acids could belong to the fossil, but sequencing by higher resolution methods would be needed for confirmation [24]. 

Antibodies raised against β-keratin have shown positive binding to demineralized thin sections of claw sheaths from the 75-million-year-old oviraptorid dinosaur, *Citipati osmolskae*, from the Djadokhta Formation of Mongolia, which showed keratinous-like microstructures [23]. Reference samples of extant emu and ostrich claw sheath were additionally studied. An in situ immunohistochemical approach combined with immunofluorescence and electron microscopy was employed that reaffirmed the previous claims that β-keratin can be preserved over millions of years. However, the available sample material from dinosaur fossils limits sequencing approaches. Yet, a targeted high-resolution mass spectrometric approach has been suggested for further studies based on sampling the regions which exhibited positive reactivity to β-keratin antibodies [23].

## 4. Conclusions 

This review provides a collection of organic compounds identified in dinosaur bone and soft tissues to date, giving insights into their chemistry and the analytical techniques used for their identification. Reports on organic compounds are increasing as more targeted sensitive analytical approaches that use less and less sample material are being developed. Organic compounds detected from dinosaurs so far comprise pigments, such as porphyrins and melanins, and proteins, including collagen type I, collagen type II and β-keratin. The analytical techniques used have been a combination of imaging using microscopy, absorption, reflectance and vibrational spectroscopy. Chemical imaging on the sample surface using time-of-flight secondary ion mass spectrometry, and more invasive techniques, namely liquid chromatography coupled with tandem mass spectrometry were also employed. Yet, even as analytical techniques become more advanced and highly sensitive, it still remains challenging to prove the endogeneity of the detected structures, especially when searching for proteins or DNA. Further development of sample preparation techniques that minimizes contamination is required. 

## Figures and Tables

**Figure 1 biology-11-00670-f001:**
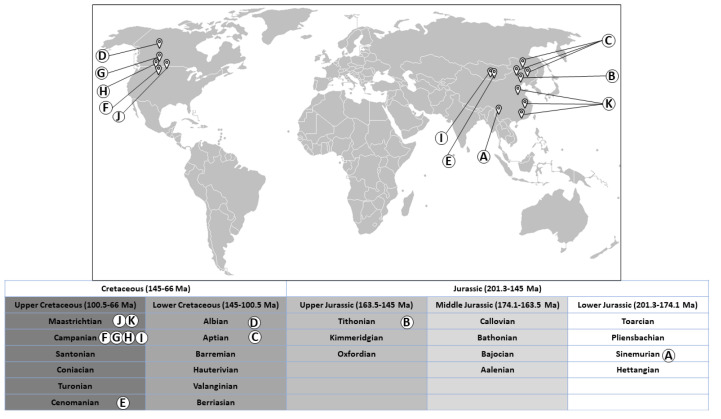
World map showing localities and age of dinosaurs in which organic compounds have been detected to date: (**A**) Dawa, Lufeng County, Yunnan Province, China [3]. (**B**) Yaolugao locality in Jianching County, western Liaoning Province, China [25]. (**C**) Dawangzhangzhi, Lingyuan City, Liaoning Province, China) and Sihetun, Beipiao City, Liaoning Province, China), and Yixian Formation, China [17,28]. (**D**) Suncor Millenium Mine, Fort McMurray, Alberta, Canada [27]. (**E**) Ukhaa Tolgod in southwestern Mongolia [24]. (**F**) Judith River Formation, eastern Montana, USA [29,30]. (**G**) Dinosaur Park Formation, Alberta, Canada [26]. (**H**) Two Medicine Formation, northern Montana, USA [31]. (**I**) Djadokhta Formation, Mongolia [23]. (**J**) Hell Creek Formation, eastern Montana, USA [18,19,20] (**K**) Chinese provinces (Henan, Jiangxi, and Guangdong) [21,22]. Concept adapted from reference [76]. The world map “BlankMap-World-IOC” by Chanheigeorge (https://commons.wikimedia.org/wiki/File:BlankMap-World-IOC.PNG, accessed on 19 March 2022) from 2008 has been used as a template onto which location markers, lines and letters were added. It is licensed under CC-BY-SA 3.0 (https://creativecommons.org/licenses/by-sa/3.0/legalcode, accessed on 19 March 2022) via Wikimedia Commons.

**Figure 2 biology-11-00670-f002:**
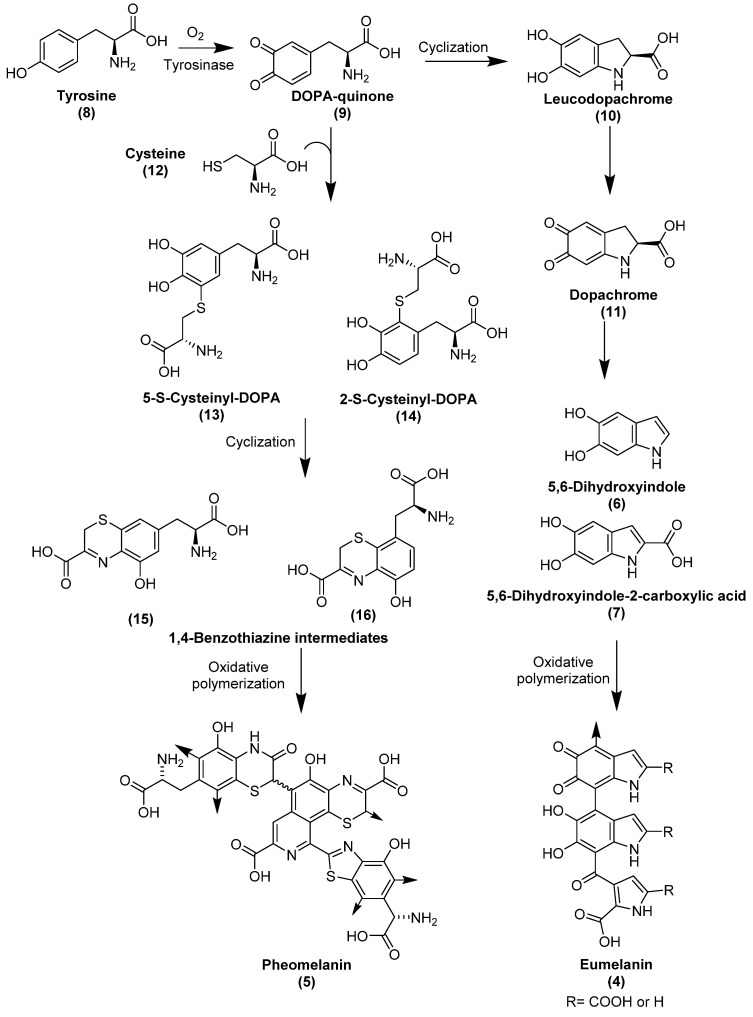
Structures and biosynthesis of eumelanin and pheomelanin. Arrows on structures (**4**) and (**5**) show points of polymer expansion [85,86].

**Figure 3 biology-11-00670-f003:**
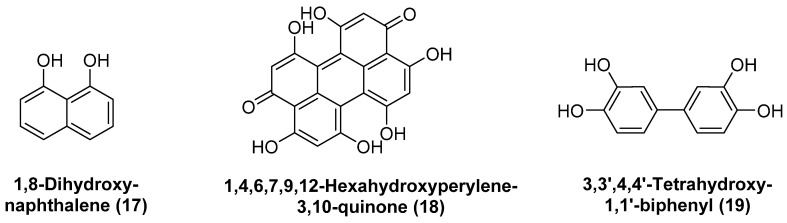
Structures of precursors involved in different types of allomelanin.

**Figure 4 biology-11-00670-f004:**
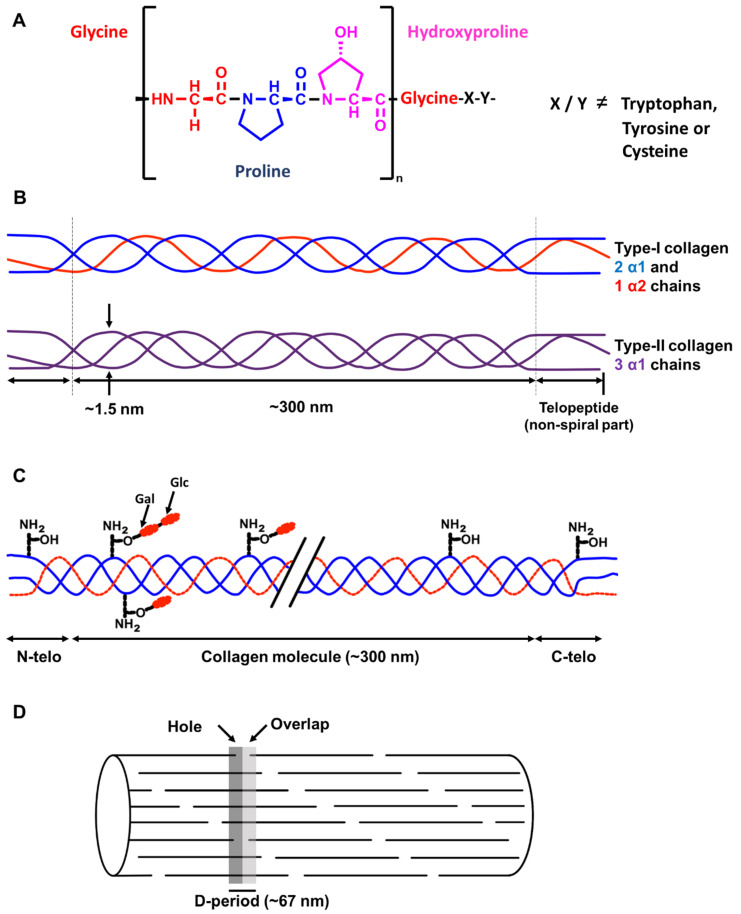
(**A**) The most common repeating sequence present in collagen types I and II. Positions X and Y can be occupied by any amino acid except tryptophan, tyrosine or cysteine. The most common amino acids in positions X and Y are proline and hydroxyproline, respectively. (**B**) Schematic representation of the triple helical structure of collagens type I and II. In collagen type I, there are two α1 chains and one α2 chain, whereas in collagen type II, there are three α1 chains. (**C**) Diagram of a collagen molecule showing the post-translational modifications that occur, which are hydroxylation of lysine residues and glycosylation of hydroxylysine by galactose and glucose. (**D**) The stacked arrangement of collagen fibers, visible under a transmission electron microscope, shows a characteristic staggered pattern known as the D-band or D-period of approximately 67 nm in periodicity. This banding is a unique feature used for identification of collagen fibers under the microscope. Adapted from [110,111].

**Figure 5 biology-11-00670-f005:**
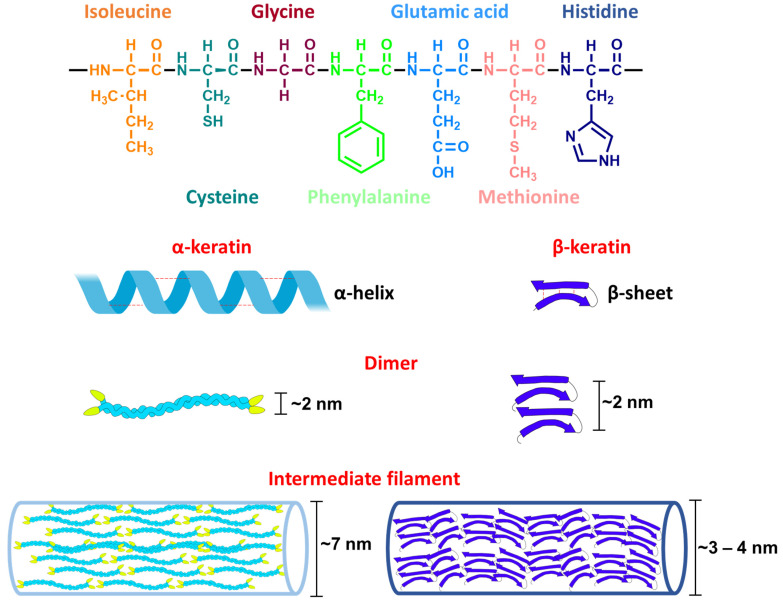
Diagram of the four different levels of keratin structure. The primary sequence of keratin is shown, including the most common amino acids present (the amino acids are L-configurated, but the stereochemistry is not shown). The secondary structure of keratins can be either an α-helix or a β-sheet, classifying them into α-keratins and β-keratins, respectively. The tertiary structures of both keratin types are heterodimers. The quaternary structure is composed of intermediate filaments, that are 7 nm in diameter for α-keratin and 3–4 nm in diameter in β-keratin. Adapted from [124].

**Table 1 biology-11-00670-t001:** Porphyrins detected in dinosaurs.

Organic Compound	Heme	Protoporphyrin IX	Biliverdin
Structure and exact mass	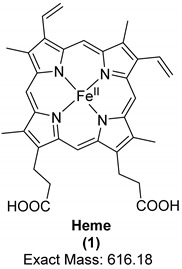	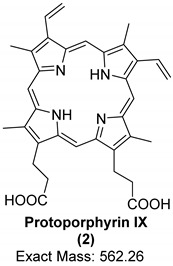	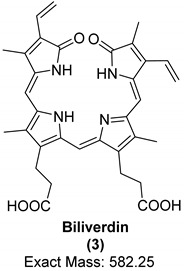
Analytical technique	HPLC-UV UV/Vis spectroscopy Raman spectroscopy	LC-ESI-q/TOF-MS Raman spectroscopy	LC-ESI-q/TOF-MS Raman spectroscopy
Dinosaur species and age	*Tyrannosaurus rex* (67 Ma)	*Heyuannia huangi* (66 Ma)	*Heyuannia huangi* (66 Ma)
Location of fossil	Hell Creek formation, eastern Montana, USA	Chinese provinces (Henan, Jiangxi, and Guangdong)	Chinese provinces (Henan, Jiangxi, and Guangdong)
Type of tissue	Extracts of trabecular bone tissues	Extract of eggshells	Extract of eggshells
Reference	[18]	[21,22]	[21,22]

**Table 2 biology-11-00670-t002:** Melanin detected in dinosaurs.

	Eumelanosomes and Pheomelanosomes	Eumelanin-like Pigmentation (Black and Yellow)	Eumelanin	Mixture of Pheomelanin and Eumelanin
Analytical technique	SEM imaging combined with EDS	Imaging with digital camera	TOF-SIMS EDS IR micro-spectroscopy	TOF-SIMS Py-GC-MS EDS
Dinosaur, location and age of fossil	*Sinosauropteryx* (125 Ma, Dawangzhangzhi, Lingyuan City, Liaoning Province, China) *Sinornithosaurus* (125 Ma, Sihetun, Beipiao City, Liaoning Province, China)	*Psittacosaurus* (125 Ma) Yixian formation in China	*Anchiornis huxleyi* (150 Ma) Yaolugao locality in Jianching County, western Liaoning, China	*Borealopelta markmitchelli* (112 Ma) Suncor Millenium Mine, Fort McMurray, Alberta, Canada
Type of tissue	Integumentary filaments from the tail	Preserved epidermal scales scattered from head to tail	Filamentous epidermal appendages (“feathers”)	Integumentary structures (epidermis and keratinized scales)
Reference	[17]	[28]	[25]	[27]

**Table 3 biology-11-00670-t003:** Collagen type I and II in the dinosaur fossil record.

Study	Collagen Type I	Analytical Technique(s)	Dinosaur Name, Location and Age	Type of Tissue	Reference
1	Amino acid fragments and peptide sequences (5 from α1 chain, 1 from α2 chain)	Immuno-histochemistry, ELISA, TOF-SIMS and LC-MS/MS	*Tyrannosaurus rex* (68 Ma) Hell Creek Formation, eastern Montana, USA	Trabecular bone	[19,20]
2	Infrared absorption bands	SR-FTIR and confocal Raman microscopy	*Lufengosaurus* (ca. 195 Ma) Dawa, Lufeng County, Yunnan Province, China	Rib bone (thin sections)	[3]
3	Amino acid fragments (alanine, arginine, glycine, and proline)	TOF-SIMS	Various Dinosauria (75 Ma) Dinosaur Park Formation, Alberta, Canada	Claw, ungual phalanx, astragalus, tibia, rib	[26]
4	Peptide sequences (6 for α1 chain, 2 for α2 chain)	Immuno-histochemistry, Western blot, ATR-IR, TOF-SIMS, and LC-MS/MS	*Brachylophosaurus canadensis* (80 Ma) Judith River Formation, eastern Montana, USA	Femur from hind limb (4 different samples)	[29]
5	Peptide sequences (6 for α1 chain, 2 for α2 chain)	Nano-LC-MS/MS and FT-ICR-MS	*Brachylophosaurus canadensis* (80 Ma) Judith River Formation, eastern Montana, USA	Femur from hind limb (4 different samples)	[30]
6	Collagen type II	Immunohisto-chemistry	*Hypacrosaurus stebingeri* (75 Ma) Two Medicine Formation, northern Montana, USA.	Calcified cartilage from supraoccipital	[31]

**Table 4 biology-11-00670-t004:** Evidence of beta-keratin in the dinosaur fossil record.

	β-Keratin and Its Amino Acid Fragments	β-Keratin Epitopes
Analytical technique(s)	TOF-SIMS Immunohistochemistry	Immunohistochemistry
Dinosaur species Location and age of fossil	*Shuvuuia deserti* (100 Ma) Ukhaa Tolgod in southwestern Mongolia	*Citipati osmolskae* (75 Ma) Djadokhta Formation of Mongolia
Type of tissue	Feather-like epidermal appendages	Original keratinous-like claw sheath
Reference	[24]	[23]

## Data Availability

Not applicable.

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
