# Peer review of "Chemistry and Analysis of Organic Compounds in Dinosaurs"

_biology, 2022, doi:10.3390/biology11050670_

Round 1

Reviewer 1 Report

Dear authors. I found your review paper appealing and well-written. I have only some minor comments/suggestions (see annotated PDF attached). 

Sincerely, 

Reviewer 2 Report

Dear authors, 

I have read your ordained overview of the state-of-art regarding the organic chemistry of dinosaur fossils. I suggest revising your manuscript by looking at the suggestions included in the attached edited manuscript file. In particular, pay special attention to the Abstract and Introduction. In addition, please do not forget to format the paper according to the MDPI formatting rules, and be sure to include Table 3 in the revised version of your paper. 

The reviewer

Reviewer 3 Report

This manuscript is so well done in its clear writing, careful scholarship, and thorough documentation, that I have very little to add or critique. There are only two minor edit changes that have to be done. The first one is located in line 306, where a punctuation mark is repeated. The other edit change is located in lines 744 and 745, instead of table 5, it should be table 4.

I recommend publication of the paper after the minor changes are done.

Author Response

Thank you. We have corrected the errors as indicated.